# An Efficient Solid-Phase Microextraction–Gas Chromatography–Mass Spectrometry Method for the Analysis of Methyl Farnesoate Released in Growth Medium by *Daphnia pulex*

**DOI:** 10.3390/molecules27238591

**Published:** 2022-12-06

**Authors:** Nicolò Riboni, Antonio Suppa, Annamaria Buschini, Federica Bianchi, Valeria Rossi, Gessica Gorbi, Maria Careri

**Affiliations:** Department of Chemistry, Life Sciences and Environmental Sustainability, University of Parma, Parco Area delle Scienze 11/A and 17/A, 43124 Parma, Italy

**Keywords:** juvenile hormone, solid-phase microextraction, gas chromatography–mass spectrometry, *Daphnia pulex*, crowding stress response, MK801

## Abstract

Methyl farnesoate (MF), a juvenile hormone, can influence phenotypic traits and stimulates male production in daphnids. MF is produced endogenously in response to stressful conditions, but it is not known whether this hormone can also be released into the environment to mediate stress signaling. In the present study, for the first time, a reliable solid-phase microextraction–gas chromatography–mass spectrometry (SPME-GC-MS) method was developed and validated for the ultra-trace analysis of MF released in growth medium by *Daphnia pulex* maintained in presence of crowding w/o MK801, a putative upstream inhibitor of MF endogenous production. Two different clonal lineages, I and S clones, which differ in the sensitivity to the stimuli leading to male production, were also compared. A detection limit of 1.3 ng/L was achieved, along with good precision and trueness, thus enabling the quantitation of MF at ultra-trace level. The achieved results demonstrated the release of MF by both clones at the 20 ng/L level in control conditions, whereas a significant decrease in the presence of crowding was assessed. As expected, a further reduction was obtained in the presence of MK801. These findings strengthen the link between environmental stimuli and the MF signaling pathway. *Daphnia pulex*, by releasing the juvenile hormone MF in the medium, could regulate population dynamics by means of an autoregulatory feedback loop that controls the intra- and extra-individual-level release of MF produced by endogenous biosynthesis.

## 1. Introduction

Methyl farnesoate (MF) is a sesquiterpenoid hormone involved in physiological and developmental processes in crustaceans [1,2,3,4]. MF plays important roles in the regulation of development [5] and stimulates male production in daphnids [6,7,8,9,10,11]. MF programs oocytes in late stages of maturation to be developed into male offspring [12,13].

Although the molecular mechanisms underlying up- and downstream signaling of MF have not yet been well-elucidated, it seems to play a leading role of conductor between external environmental stimuli and the endogenous male developmental pathway. A possible link between MF signaling and sex determining could involve glutamate receptors, as published by the research groups of Toyota and Camp [11,14]. Recently, our research group demonstrated the influence of MF on *D. pulex* life-history traits [15]. The achieved results suggested that the effects of high population density/food shortage may be mediated by the endogenous biosynthesis of MF [15].

MF has been detected in tissues and organs of both crustacean species and insects. Quantification of MF in hemolymph of *Nauphoeta cinerea* by liquid chromatography coupled to UV detection (HPLC-UV) was proposed for the first time by Borst and Tsukimura in 1991 [16]. Subsequently, Rotllant and coworkers quantitated MF in *Nephrops norvegicus* [17], whereas the effect of stress response on MF concentration in *Carcinus maenas* and in vitro secretion by mandibular organs of *Oziotelphusa senex senex* was studied by the research groups of Lovett [18] and Nagaraju [19], respectively. Recently, ultra-high performance liquid chromatography–tandem mass spectrometry (UHPLC-MS/MS) was proposed for the identification and quantification of the five juvenile hormone homologs and MF in different species of Diptera, Lepidoptera, Heteroptera, and Hymenoptera [20]. Requiring no sample treatment or chromatographic separation, real-time mass spectrometry (DART-MS) was also tested for the detection of MF: in this case, standard solutions were used, obtaining a detection limit of 650 fmol/µL [21]. However, among the available instrumental techniques, the most commonly applied for both identification and quantitation of MF is gas chromatography–mass spectrometry (GC-MS) due to its high sensitivity and selectivity. Both electron ionization (EI) [17,22,23,24,25] and chemical ionization (CI) [26,27,28,29] have been proposed for MF ionization. GC-MS/MS has been also proposed for the analysis of juvenile hormones and MF both in vitro and in vivo [30].

As for sample treatment, in most of the studies dealing with the determination of MF in the hemolymph of insects and crustaceans, a liquid-liquid extraction (LLE) is applied, using hexane as extraction solvent [16,17,18,19,20,22,23,24,25,27,28,30]. To overcome the drawbacks of the classical LLE methods, i.e., the use of large volumes of toxic and expensive solvents, the difficulty of automation and long extraction times, other sample preparation methods, namely micro-solid phase extraction (MSPE) and matrix-solid phase dispersion (MSPD), have been proposed [26,29]. However, the use of organic solvents and the lack of automation are still the major drawback of these procedures. Solid-phase microextraction (SPME), a versatile, automatable, and easy-to-use extraction technique, has been proposed as a valuable alternative to most laborious extractions [31] for a wide range of compounds both in vitro and in vivo [32,33,34,35,36] and proven to be suitable for the analysis of hormones in both insects and crustaceans [37,38,39].

For the first time in the present study, a SPME-GC-MS method is proposed for the analysis of MF in water conditioned by daphnids maintained under selected environmental conditions, i.e., mimed crowding with or without the presence of MK801, a putative upstream inhibitor of endogenous production of MF [10]. Our hypothesis is that Daphnia might release juvenile hormone MF in the growth medium, thus potentially affecting and regulating population dynamics.

## 2. Results and Discussion

### 2.1. SPME Optimization

To evaluate MF releasing by *D. pulex* maintained under stress conditions, a SPME-GC-MS method was developed and validated. Due to its high pre-concentration factor, a microextraction technique, namely SPME, was used for the determination of MF at ultra-trace level in the growth medium. 

Considering that the nature of the SPME coating plays a pivotal role on the method performance by affecting sensitivity, selectivity, and repeatability [40,41], preliminary experiments were carried out to select the most appropriate SPME coating. The extraction capabilities of two commercially available fibers, namely 1 cm–30 μm PDMS and 1 cm–50/30 μm DVB-CAR-PDMS, were compared in terms of GC-MS responses showing the superior performance of the DVB-CAR-PDMS fiber, with responses eight times higher than those achieved by using the PDMS fiber, namely DVB-CAR-PDMS (17.6 ± 1.5) × 10^5^ vs. PDMS (22.2 ± 1.6) × 10^4^.

The SPME conditions were optimized in terms of extraction temperature (T) and extraction time (t) by means of a Box–Wilson composite face-centered design (CCF) design. Considering the need of a uniform and repeatable heating of the extraction vial, the effects of extraction temperature were investigated in the 40–80 °C range. As for the extraction time, the 10–50 min range was chosen in order to favor a good adsorption of MF onto the coating while avoiding prolonged extractions.

The following regression model was calculated:y = 228,000 (±11,000) + 119,000 (±15,000)t + 76,000 (±15,000)T.

Neither quadratic effect nor interaction proved to be significant (*p* > 0.05). Both extraction temperature and time resulted to be significant with positive coefficients: as already stated in previous studies, the use of high extraction temperatures increases both the overall extraction kinetic and MF diffusion coefficient [42,43,44], whereas a prolonged extraction time is required for a better adsorption of MF onto the SPME fiber.

Therefore, SPME extraction was carried out using 1 cm–50/30 μm DVB-CAR-PDMS fiber at 80 °C for 50 min.

### 2.2. SPME-GC-MS Method Validation

Method validation was performed under the optimized experimental conditions. Detection and quantitation limits of 1.3 and 4.3 ng/L, respectively, were achieved, thus demonstrating the capability of detecting MF at ultra-trace levels. Method linearity y = (336.3 ± 6.5) × 10^−5^ was assessed in the LOQ-100 ng/L range; a good precision in terms of both repeatability and intermediate precision, with RSDs always lower than 14%, was also obtained (Table 1). As for intermediate precision, ANOVA showed that the mean responses were not significantly different over time (*p* > 0.05). Recovery rates in the 90.5 ± 5.9–105 ± 13% (n = 10) range were calculated (Table 1), thus assessing the trueness of the developed method. Matrix effect was evaluated by comparing the slope of the calibration curve obtained using standard solutions with that achieved by applying the standard addition method. No significant difference was observed (*p* > 0.05). An extraction efficiency of 7% was calculated: this finding is not surprising since SPME is not an exhaustive extraction technique, and equilibrium was not reached at the optimal conditions. Finally, method selectivity was demonstrated: no change in the ion intensity ratio of both qualifiers and quantifier ions of MF was observed. 

As shown in Table 2, the developed method is characterized by the lowest LOD compared to other studies reported in the literature [16,17,20,21,22,29,30]. The achievement of low detection and quantitation limits is crucial to assess the release of MF in the growth medium under different environmental conditions since lower concentration levels than those observed in tissues and hemolymph were expected. 

### 2.3. Release of Methyl Farnesoate under Stress Conditions

Life-history traits analysis showed no significant difference in age at first reproduction between treatments and between clones. Both clones showed a 15% reduction in body size and in the number of neonates per female in crowding and MK801 treatments with respect to control. As expected, significant induction of male and ephippia production was observed only in the clone S in crowded conditions [15].

MF was detected in the growth medium for all treatments (Figure 1), but no significant difference between clones was observed (Figure 2). 

As previously reported, compared to clone S, clone I does not have the ability to produce males when not exposed to MF, but this difference has not undergone genetic alterations involving the endogenous biosynthesis of MF [15]. 

The highest MF concentration was observed in controls, whereas a significant, progressive reduction in crowding and MK801 treatments was recorded.

While endogenous MF was produced under control conditions and released in the medium, its production and/or releasing was reduced in crowding condition. A further decrease of MF in the medium was observed in presence of MK801 inhibitor, probably due to a concomitant alteration of neuroendocrine signals influencing MF biosynthesis [10]. Previous studies indicated the presence of pheromones in certain aquatic crustaceans [45], but so far, to our knowledge, no evidence was found for the presence and release of sexual pheromones in cladocerans. 

Our findings suggest that both clones naturally produce MF by endogenous biosynthesis and release it in the water medium. A further significant reduction of MF in the aqueous medium can be ascribed to the synergic effect of both crowding and MK801 inhibitor. This result should confirm the linkage between MF signaling and environmental stimuli such as crowding [14]. Under control conditions, the concentration of endogenous MF released in the water should act as a chemically mediated density-dependent mechanism that activates male production [46].

## 3. Materials and Methods

### 3.1. Chemicals and Materials

Methanol (>99% purity), methyl decanoate (>99% purity), and MK801 [(5S,10R)-(+)-5-Methyl-10,11-dihydro-5H-dibenzo[a,d]cyclohepten-5,10-imine maleate] (≥98% purity) were purchased from Merck (Milan, Italy). (E,E)-methyl farnesoate was acquired from Echelon Biosciences (Salt Lake City, UT, USA).

Polydimethylsiloxane (PDMS, 30 μm, 1 cm) and divinylbenzene/carboxen/polydimethylsiloxane (DVB-CAR-PDMS, 50/30 μm, 1 cm) fibers were purchased from Supelco (Bellefonte, PA, USA).

### 3.2. GC-MS Analysis

GC-MS analyses were performed by using a HP 6890 Series Plus gas chromatograph (Agilent Technologies, Palo Alto, CA, USA) equipped with a MSD 5973 mass spectrometer (Agilent Technologies, Palo Alto, CA, USA). The following operating conditions were applied: carrier gas—helium—at a constant flow rate of 1.2 mL/min; GC injector: splitless mode at 260 °C; chromatographic separation on a Rxi-5Sil MS capillary column (30 m × 0.25 mm i.d., 0.25 μm film thickness; Restek, Bellafonte, USA), applying the following temperature program: 90 °C hold for 2 min, 15 °C/min to 180 °C, 20 °C/min to 270 °C; transfer line and ion source at 280 °C and 150 °C, respectively. Full-scan EI data were acquired to determine appropriate m/z ion ratios for time scheduled monitoring using the following conditions: ionization energy: 70 eV; mass range: 50–350 amu; scan time: 3 scan/s; electron multiplier voltage: 1000 V, solvent delay 2.00 min. All the analyses were carried out by operating in selected ion monitoring mode (SIM), dwell time 30 ms, monitoring the following ions: m/z **74**, 143 and 186 for methyl decanoate; m/z **69**, 114 and 250 for methyl farnesoate (ions used for quantitation in bold). Signal acquisition and data handling were performed using the HP Chemstation (Agilent Technologies).

### 3.3. Method Optimization

SPME analyses were performed using a PAL COMBI-xt autosampler (CTC Analytics AG, Zwingen, Switzerland) operating in direct immersion (DI) mode. Prior to use, the fibers were conditioned in the GC injection port as indicated by the producer. The analyses were performed using 9.5 mL of aqueous solutions in 10 mL sealed glass vials. 

The SPME-GC-MS method was optimized considering fiber coating, extraction temperature and time. Firstly, two different fibers were tested, namely 1 cm–30 μm PDMS and 1 cm–50/30 μm DVB-CAR-PDMS, to select the coating showing the highest extraction capacity: for this purpose, 5 µg/L aqueous solution of MF was incubated at 60 °C for 7 min and extracted at the same temperature for 30 min in direct immersion mode under an agitation speed of 250 rpm. The extraction performances were compared in terms of GC-MS responses: Shapiro–Wilk and Bartlett tests were carried out to assess the normal distribution and the homoscedasticity of data, respectively, then a *t*-test was performed to determine if there was a significant difference between the coatings.

As for extraction temperature and time, in order to investigate the effects and the presence of significant interactions, the optimization was performed using a CCF design. The experimental domain was explored in the 40–80 °C and in the 10–50 min range. NW containing MF at 1 µg/L was extracted by DI-SPME. The experimental error was evaluated by performing four replicates at the center of the experimental domain. The presence of relevant quadratic effects was assessed running a F-test by comparing the experimental and calculated responses at the center of the experimental domain [42,47].

The significance of the main factors, quadratic effects, and their interactions was evaluated. The regression models were obtained by a forward search stepwise variable algorithm (*p* to remove 0.05) by using the statistical package SPSS Statistics v.23.0 (IBM, Milan, Italy).

### 3.4. Method Validation

Method validation was performed according to EURACHEM guidelines [48] under the optimized conditions using uncontaminated natural water (NW) as blank matrix. Detection (yD) and quantitation (yQ) limits were expressed as signals based on the mean blank (xb) and the standard deviation of blank responses (sb) as follows: yD = xb + 3 sb and yQ = xb + 10 sb. The value of xb and sb were calculated performing ten blank measurements. Detection and quantitation limits (LOD and LOQ, respectively) were obtained by projection of the corresponding signals yD and yQ through a calibration plot y = f(x) onto the concentration axis. For quantitative purposes, a proper calibration curve (six concentration levels, three replicated measurements for each level) was built between the LOQ and 100 ng/L using methyl decanoate as internal standard at the concentration of 100 ng/L. Homoscedasticity was verified by applying the Bartlett test. Lack of fit and Mandel’s fitting test were also performed to assess the goodness of fit and linearity. The significance of the intercept (significance level 5%) was established by running a Student’s *t*-test. Repeatability and intermediate precision were calculated in terms of relative standard deviations (RSD%) at three concentration levels (LOQ, 30 and 80 ng/L) by performing six replicated measurements per level. Intermediate precision was estimated over three days verifying homoscedasticity of data and performing the analysis of variance (ANOVA) at the confidence level of 95%. Trueness was calculated in terms of recovery rate (RR%) as follows: RR% = c1/c2∙100, where c1 is the measured concentration, and c2 is the concentration obtained by spiking the blank sample at LOQ, 30 and 80 ng/L. Recovery rate values were assessed by performing ten replicated measurements per level at the same levels used for repeatability. Extraction efficiency was evaluated under the optimized conditions by calculating the amount of MF onto the fiber coating to that present in a spiked water sample at the concentration of 100 ng/L. Selectivity was evaluated by analyzing water samples and verifying the absence of interfering compounds.

### 3.5. Daphnia pulex Culturing

Females of *D. pulex* pertaining to two genetically different clones (Italy, I, and Sedlec, S), previously selected according to Suppa and coworkers [15], were used. Females of both clones were bred in NW at 20 ± 1 °C with photoperiod of 14:10 (L:D) hours, fed twice a week with the unicellular green alga *Pseudokirchneriella subcapitata* (at a density of 1.5 × 10^5^ cells/mL) and the yeast *Saccharomyces cerevisiae* (at a density of 1.5 × 10^5^ cells/mL), and maintained at a population density of 20 females per clone in 600 mL of the culture medium that was renewed twice a week.

Life-table experiments (lasting 21 days) were performed under selected environmental conditions (population density/food availability) able to elicit diversified molecular and phenotypic responses in the two clones (clones I and S) [15,49]. The effect of the following treatments were evaluated: (1) control treatment—high food availability (1.5 × 10^5^ cells/mL of *P. subcapitata* and of *S. cerevisiae*) and low daphnid density (1 individual/50 mL) that represents the optimal condition; (2) crowding treatment—high food density (1.5 × 10^5^ cells/mL of *P. subcapitata* and of *S. cerevisiae*) and high daphnid density (1 individual/10 mL) that constitutes a situation of food shortage due to crowding; and (3) crowding + MK801 treatment—high food density (1.5 × 10^5^ cells/mL of *P. subcapitata* and of *S. cerevisiae*) and high daphnid density (1 individual/10 mL) in presence of MK801 [(5S,10R)-(+)-5-Methyl-10,11-dihydro-5H-dibenzo[a,d]cyclohepten-5,10-imine maleate], a non-competitive, high-affinity, open-channel blocker of the N-methyl-D-aspartate receptors. MK801 is a putative upstream inhibitor of MF endogenous production and should overcome the effect of food shortage due to crowding.

Ten juveniles (age <24 h) per clone per treatment (3 treatments × 2 clones × 10 replicates = 60 females) were reared individually in beakers containing 50 mL of culture medium (control treatment) or in six-well plates (one juvenile per well in 10 mL medium, crowding treatments). Daphnids were fed with a mixture of *P. subcapitata* and *S. cerevisiae*, each at the final concentration of 1.5 × 105 cells/mL. In each treatment, NW medium and food were renewed three times a week. 

Body size (as carapace length from the top of the head to the base of the tail spine) was measured on the 8th day of treatment; age at maturity (the age at which the first eggs are deposited into the brood chamber) was checked daily; fecundity (total number of alive offspring per female) was recorded three times a week. Juveniles were classified according to their gender using a CLS 50× microscope (Leica CLS 50× microscope; Leica, Mannheim, Germany). 

At the end of the experiment, daphnids were separated by mesh filtration, and the water samples (one vial per clone per treatment) in which the adult females were maintained for the last 48 h were stored at −20 °C.

All water samples were then submitted to SPME-GC-MS analysis. Shapiro–Wilk and Bartlett tests were performed to assess the normal distribution and the homoscedasticity of data. Two-way ANOVA followed by Bonferroni’s comparison test was performed to assess the presence of significant differences between the treatments and the two different clones. Differences were considered significant when *p* < 0.05.

## 4. Conclusions

The SPME-GC-MS method developed and validated in this study was applied for the first time, for the analysis of MF released by daphnids in growth medium, demonstrating adequate performance in terms of sensitivity, trueness, and precision. In particular, the LOQ value at the low ng/L level allowed the quantitation of MF released under stress conditions by *D. pulex*. The achieved results suggest the linkage between environmental stimuli and MF signaling pathway. Despite their constitutive genetic differences in propensity to produce male and ephippium, the analyzed clones showed no difference in the production and release of MF in water. These results suggest that the response cascade is qualitatively identical at both phenotypic and transcriptional levels in the MF biosynthesis. This is the first time, to our knowledge, that MF has been searched and detected in media conditioned by *Daphnia pulex*. Our results will need to be confirmed by further studies in which crowding is modulated by the presence of other organisms of the same species.

## Figures and Tables

**Figure 1 molecules-27-08591-f001:**
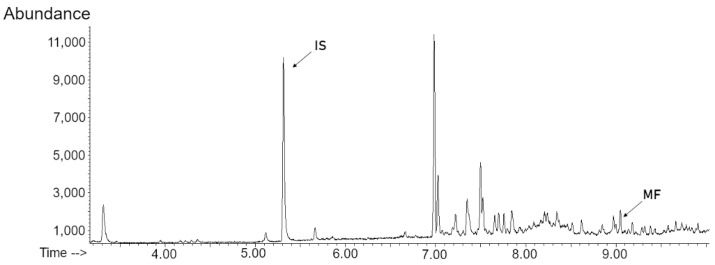
SPME-GC-(SIM)-MS chromatogram of a growth medium of *D. pulex* (clone I) under crowding conditions.

**Figure 2 molecules-27-08591-f002:**
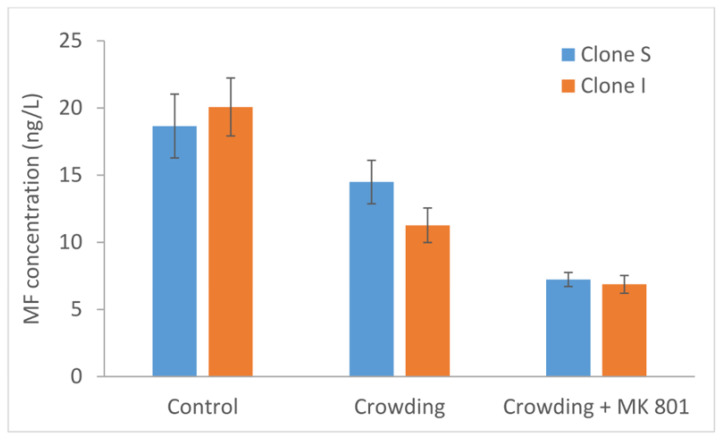
Concentration of MF released in the growth medium by clones S and I of *D. pulex* under control, crowding, and crowding + MK801 conditions.

**Table 1 molecules-27-08591-t001:** Precision and trueness of the SPME-GC-MS method.

	Concentration Level (ng/L)		
	LOQ	30	80
Repeatability RSD%	13	9.5	7.5
Intermediate Precision RSD%	14	11	7.8
Trueness RSD% (±st.dev.)	105 ± 13	98.1 ± 9.0	90.5 ± 5.9

calibration curve y = a + bx.

**Table 2 molecules-27-08591-t002:** LOD values reported in previously published studies for the determination of MF.

Article	LOD (µg/L)	Extraction Method	Analytical Technique	Organism	Matrix
This study	0.0013	SPME	GC-(EI)MS	*Daphnia pulex*	Growth medium
[16]	0.25	LLE	HPLC-UV	*Homarus americanus, Libinia emarginata, Carcinus maenas*	Hemolymph
[22]	50	LLE	GC-(EI)MS	*Cancer pagurus*	Hemolymph
[23]	100	LLE	GC-(EI)MS	*Balanus amphitrite*	Dry-blotted cyprids
[24]	0.3	LLE + preparative HPLC	GC-(EI)MS	*Procambarus clarkii*	Hemolymph
[21]	160	-	DART-MS	*-*	-
[30]	1	LLE	GC-MS/MS	*Diploptera punctata*	Hemolymph and corpora allata
[29]	0.5	MSPD	GC-(CI)MS	*Gammarus locusta, Artemia franciscana, Apis mellifera*	Whole arthropod
[20]	0.007	LLE	UHPLC-MS/MS	*Aedes aegypti, Sarcophaga bullata, Oncopeltus fasciatus, Manduca sexta, Bombyx mori, Drosophila melanogaster, Megalopta genalis, Anopheles albimanus, Dipetalogaster maxima*	Hemolymph, corpora allata–corpora cardiaca and brain and head capsule

## Data Availability

Not applicable.

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
