# Peer review of "An Efficient Solid-Phase Microextraction–Gas Chromatography–Mass Spectrometry Method for the Analysis of Methyl Farnesoate Released in Growth Medium by Daphnia pulex"

_molecules, 2022, doi:10.3390/molecules27238591_

Round 1

Reviewer 2 Report

The article entitled "An efficient solid-phase microextraction-gas chromatography-mass spectrometry method for the analysis of methyl farnesoate released in growth medium by Daphnia pulex " presents an interesting study that develops and validates the method for the analysis of methyl farnesoate released in growth media by Daphnia pulex.  As indicated in its conclusions, the results of this study will be very useful for the study of this analyte in Daphnia pulex growth media. The results obtained suggest the relationship between environmental stresses and the MF signaling pathway. The clones analyzed, despite their genetic differences, showed no differences in the production and release of MF in water.

I indicate below a series of aspects that need to be clarified in order to be published.

1.- Lines 2, 3, 4 and 5 of the third paragraph of the introduction should be deleted as they do not provide new information. In the following lines they are commented again.

2.- In the second paragraph of section 2.1. I do not understand the 8-fold increase in the response with the data indicated by the authors.

3.- In the second line of the third paragraph of section 2.1. As this is the first time that "CCF" appears in the text, you should indicate what the initials stand for.

4.- In the fourth line of the third paragraph of section 2.1. Because the maximum time selected to make the design is 50 minutes.

5.- In the fourth paragraph of section 2.1, indicate the p-values corresponding to the F-test to evaluate the significance of the quadratic terms, since they indicate that they have done the test.

6.- In the first paragraph of section 4.2. In my version of the manuscript there is no fragment in bold, so I can not know what ion they use to quantify.

7.- In the second line of section 4.3. Because they used the direct immersion mode and not another one.

8.- In the second line of section 4.4. How do you ensure that this water is free of contaminants?

9.- The last two lines of section 4.4. indicate that selectivity studies have been carried out, but the selectivity results obtained are not indicated in the corresponding section.

Round 2

Reviewer 1 Report

After revision, in the corrected and edited version the authors have complied with the main recommendations amd remarks by adding additional texts, explanations and figures  in the relevant paragraphs.